# No Impact of Biocontrol Agent’s Predation Cues on Development Time or Size of Surviving *Aedes albopictus* under Optimal Nutritional Availability

**DOI:** 10.3390/insects13020155

**Published:** 2022-01-31

**Authors:** Marie C. Russell, Lauren J. Cator

**Affiliations:** Department of Life Sciences, Silwood Park Campus, Imperial College London, Ascot SL5 7PY, UK; l.cator@imperial.ac.uk

**Keywords:** predator–prey interactions, sublethal effects, vector traits

## Abstract

**Simple Summary:**

*Aedes albopictus* is a highly invasive species of mosquito that can infect humans with chikungunya, dengue, yellow fever, and Zika. Within the next few decades, this mosquito species is predicted to invade South East England. Cyclopoid copepods are small crustaceans that have previously been used as biocontrol agents due to their high efficiency at killing small early instar mosquito larvae. We assessed the effect of *Megacyclops viridis*, a copepod species local to South East England, on the survival and traits of mosquito larvae exposed to these predators during the larger late instar stage. Our experiment was designed to measure the impact of copepod predation on both the development time and adult body size of *Ae. albopictus*. These traits can shape mosquito population dynamics and disease transmission. While we found that copepod attacks cause a small increase in late instar mortality, our methods did not detect a significant difference in either development time or size between the predator and control treatments. The lack of a strong sublethal effect on these traits supports the use of *M. viridis* copepods as biocontrol agents against *Ae. albopictus* in the UK. This information may be useful for guiding public health measures that aim to prevent outbreaks of mosquito-borne disease.

**Abstract:**

Cyclopoid copepods have been applied successfully to limit populations of highly invasive *Aedes albopictus* mosquitoes that can transmit diseases of public health importance. However, there is concern that changes in certain mosquito traits, induced by exposure to copepod predation, might increase the risk of disease transmission. In this study, third instar *Ae. albopictus* larvae (focal individuals) were exposed to *Megacyclops viridis* predator cues associated with both the consumption of newly hatched mosquito larvae and attacks on focal individuals. The number of newly hatched larvae surrounding each focal larva was held constant to control for density effects on size, and the focal individual’s day of pupation and wing length were recorded for each replicate. Exposing late instar *Ae. albopictus* to predation decreased their chances of surviving to adulthood, and three focal larvae that died in the predator treatment showed signs of melanisation, indicative of wounding. Among surviving focal *Ae. albopictus*, no significant difference in either pupation day or wing length was observed due to copepod predation. The absence of significant sublethal impacts from *M. viridis* copepod predation on surviving later stage larvae in this analysis supports the use of *M. viridis* as a biocontrol agent against *Ae. albopictus*.

## 1. Introduction

*Aedes albopictus* is an important vector of dengue, chikungunya, yellow fever, and Zika [1]. This species is highly invasive, in part due to its ability to lay desiccation-resistant eggs that can be transported across long distances, often by the shipment of used tires [2,3,4,5,6]. The northern limits of the global range of *Ae. albopictus* include North American populations in New York and Connecticut, USA [7], and European populations throughout Italy and France [8].

Several different methods have been proposed for controlling aedine mosquitoes in Europe, including chemical, genetic, and biological techniques [9]. Although space spraying with pyrethroid insecticide was effective against *Ae. albopictus* in Catalonia, this control strategy would require regular monitoring for insecticide resistance [10]. The sterile insect technique (SIT) was tested on *Ae. albopictus* mosquitoes from Italy, but it did not significantly reduce their population due to detrimental effects on mating competitiveness among irradiated males [11]. Cyclopoid copepods were used successfully as biocontrol agents against mosquito larvae in the US [12], Australia [13], Vietnam [14], and Italy [15]. Copepods are an especially convenient type of biocontrol because they are small enough to be distributed through a backpack sprayer [16]. However, exposure to predators over multiple generations can result in greater mosquito population performance, as measured by a composite index of performance (*r*’) that takes mosquito wing length into account [17,18].

Copepod predators are much more effective at killing first and second instars than third and fourth instar *Ae. albopictus* [19]. Thus, after an application of copepod predators to *Ae. albopictus* larval habitats, it is likely that any immatures that were third and fourth instar larvae at the time of application would still emerge as adults. Because *Ae. albopictus* eggs do not hatch all at once [20], these individuals are likely to withstand both unsuccessful copepod attacks and exposure to the deaths of surrounding smaller conspecifics. Such sublethal interactions between predator and prey species can strongly impact prey traits [21,22].

In aquatic ecosystems, sublethal effects are governed by olfactory cues, including predator kairomones and chemical alarm cues from the prey [23]. For example, *Culex pipiens* larvae significantly reduce their movement when exposed to conspecifics that have been killed by notonectid predators [24]. In *Cx. restuans*, both freezing (hanging at the water’s surface or drifting in the water column) and fleeing (moving for nearly a full minute without resting) behaviours were observed as responses to conspecific alarm cues [25]. *Cx. restuans* larvae have also learned to recognize a salamander predator’s odour as a threat, after experiencing the predator’s odour paired with conspecific alarm cues [26]. When *Ae. triseriatus* larvae were exposed to chemical cues of *Toxorhynchites rutilus* predation until adult emergence, females exhibited significantly shorter development times under high nutrient availability [27].

Adult body size is an important vector trait that can be affected by predation. For example, exposure to a backswimmer predator, *Anisops*
*jaczewskii*, reduced the size of *Anopheles coluzzii*, one of the main vectors of the human malaria parasite *Plasmodium falciparum* in Africa [28]. Similarly, larval exposure to chemical cues from predatory fish *Hypseleotris galii* reduced the adult size of *Ae. notoscriptus*, the main vector of canine heartworm *Dirofilaria immitis* in Australia [29]. However, in comparison to how other mosquito species respond to predators, studies on *Ae. albopictus* indicate that this species may be less sensitive to predator cues [30,31].

As an invader, *Ae. albopictus* is less likely to share an evolutionary history with the predators that it encounters. A lack of a shared evolutionary history between predator and prey species can weaken non-consumptive predator effects in arthropods [32]. Previous work showed that second instar *Ae. albopictus* reduce their movement in response to cues from the act of predation by *Corethrella appendiculata*, which likely includes dead conspecifics and predator faeces, but not in response to cues from a non-feeding predator [33]. However, larger fourth-instar *Ae. albopictus* are less vulnerable to predation and show no behavioural response to cues from *C. appendiculata*, even to cues from the act of predation [34].

If copepods are to be used as biocontrol agents, it is important to understand how predation may alter the adult traits of larvae that are too large to be consumed at the time of application. The body size of aedine mosquitoes can be altered by predation cues [29,35], and changes in size can have important and complex consequences for population dynamics and disease transmission. In *Ae. albopictus*, larger adults have greater reproductive success [36,37]. Larger males produce more sperm [37], and in larger females, more spermathecae contain sperm [38]. A larger female body size consistently correlates with higher fecundity when measured as the number of mature follicles in the ovaries [39] or as the number of eggs laid [36]. Thus, larger emerging *Ae. albopictus* could lead to increased mosquito abundance.

In addition, larger *Ae. albopictus* adults have displayed significantly longer median lifespans when controlling for differences due to sex and diet [40]. Vector lifespan is a particularly important trait for predicting disease transmission because older vectors have had more opportunities for exposure to pathogens and are more likely to be infectious due to surviving longer than a pathogen’s extrinsic incubation period [41]. However, small *Ae. albopictus* females are more likely, relative to large females, to become infected with dengue and disseminate the virus [42]. Small females are also more likely than large females to take multiple bloodmeals within a single gonotrophic cycle [43]. This higher contact frequency with hosts could increase the risk for disease transmission by small *Ae. albopictus* vectors.

Previous studies showed an increase in *Ae. albopictus* adult size after exposure to predation, but these studies were not designed to control for the greater per capita nutrition or decreased intraspecific competition that often occur when the population density has been significantly lowered due to successful predation [30,44,45,46]. The phenomenon of increasing animal body size with decreasing population density has been documented across taxa [47]. However, impacts on the feeding behaviour of insect prey have been observed in response to predator-associated cues alone [48]. This study tests whether *Ae. albopictus* that have been exposed to predator-associated cues during the later larval stages experience sublethal effects on development time or adult size, independent of density effects. Due to the suitability of some areas in South East England for *Ae. albopictus* populations [49], we used *Megacyclops viridis*, a likely copepod species for future biocontrol applications [50], collected from Longside Lake in Egham, UK, and *Ae. albopictus* that were originally collected in Montpellier, France. Our results show that, while cyclopoid copepod predation by *M. viridis* significantly increases the mortality of late instar *Ae. albopictus*, related predation cues do not significantly change the development time or adult size of those late instar *Ae. albopictus* that survive, assuming optimal nutritional availability.

## 2. Materials and Methods

### 2.1. Local Copepod Collections

We collected 130 adult female *M. viridis* copepods during the third week of September 2019 from the edge of Longside Lake in Egham, Surrey, UK (51°24.298′ N, 00°32.599′ W). Copepods were identified as *M. viridis* (Jurine, 1820) by morphology. The copepods were kept in ten 1 L containers, each holding approximately 500 mL of spring water (Highland Spring, Blackford, UK) at a 12:12 light/dark cycle and 20 ± 1 °C. *Paramecium caudatum* were provided ad libitum as food for the copepods, and boiled wheat seeds were added to the containers to provide a food source for the ciliates [51].

### 2.2. Temperate Ae. albopictus Colony Care

A colony of *Ae. albopictus* mosquitoes (original collection Montpellier, France, 2016, obtained through Infravec2) was maintained at 27 ± 1 °C, 70% relative humidity, and a 12:12 light/dark cycle. Larvae were fed fish food (Cichlid Gold Hikari^®^, Himeji-shi, Japan), and adults were given 10% sucrose solution and horse blood administered through a membrane feeding system (Hemotek^®^, Blackburn, UK). *Ae. albopictus* eggs were collected from the colony on filter papers and stored in plastic bags containing damp paper towels to maintain humidity.

### 2.3. Experimental Procedure

On the morning of the tenth day after the last *M. viridis* had been collected from the field, *Ae. albopictus* larvae were hatched over a 3 h period at 27 ± 1 °C. Hatching temperature was kept high, relative to the temperature of the experiment (20 ± 1 °C), to maximise the yield of larvae over a semi-synchronous period. Stored egg papers were submerged in 3 mg/L nutrient broth solution (Sigma-Aldrich © 70122 Nutrient Broth No. 1), and oxygen was displaced by vacuum suction for 30 min. Immediately following oxygen displacement, ground fish food (Cichlid Gold Hikari^®^, Himeji-shi, Japan) was added ad libitum. After three hours, 500 larvae were counted and placed into spring water to dilute residual food from the hatching media. These focal larvae were then placed in a 1 L container with 600 mL of spring water and twelve pellets (each 50 mg) of fish food at 20 ± 1 °C. Water and fish food were changed every other day.

The focal larvae were held at a constant temperature of 20 °C because a previous median regression showed that summer temperatures in South East England rose from 14.9 to 17.0 °C between 1971 and 1997 [52]. This warming trend is very likely to continue, with the London climate projected to resemble that of present-day Barcelona by 2050 [53]. The average maximal summer temperature for the Greater London area from 1976 to 2003 was 22.3 °C [54]; therefore, 20 °C is within the range of realistic summer temperatures to be experienced in South East England during the next few decades.

On the sixth day of focal larvae development, 70 female *M. viridis* copepods were each placed in a Petri dish (diameter: 50 mm, height: 20.3 mm) holding 20 mL of spring water to begin a 24 h starvation period. A second set of *Ae. albopictus* eggs were hatched according to the previously described method, except that the hatch was held at 27 ± 1 °C for 18 h.

On the seventh day of focal larvae development, 140 third instar focal *Ae. albopictus* larvae were each placed in a Petri dish holding 20 mL of spring water, a 50 mg pellet of fish food, and four first instar larvae from the hatch that was started on the previous day. A subset of 30 third-instar focal larvae were preserved in 80% ethanol for head capsule width measurements [55]. Seventy starved *M. viridis* copepods were then introduced to 70 out of the 140 Petri dishes (Figure 1).

Each day, the number of surviving first or second instar larvae in each of the 140 Petri dishes was recorded, surviving first or second instars were removed, and four new first instars from the 18 h hatch started on the previous day were added to each replicate (Figure 1). The status (dead or alive) of each focal larva was recorded daily, and in predator treatment replicates, the status of the copepod was also recorded. In the case of a focal larva death, the larva was preserved in 80% ethanol, and that replicate was removed from further observation. In the case of a copepod death, the copepod was preserved in 80% ethanol, and a new adult female *M. viridis* copepod from September field collections was randomly chosen to replace it.

Pupation among focal individuals was recorded each day at 18:00. Pupae were transferred into 10 mL of spring water in a graduated cylinder with a mesh cover for emergence. Emerged adults were frozen at −20 °C. Wings were removed and measured as a proxy for body size [56].

### 2.4. Data Analysis

All analyses were completed in R version 3.4.2 [57]. Welch two-sample t-tests for samples of unequal variance were used to compare the survival percentage of first or second instar larvae between absent and present copepod treatments. The possibility of a difference in the proportion of focal larvae emerging as adults based on copepod presence was examined using a Pearson’s chi-squared test without Yates’ continuity correction. Two Kruskal–Wallis tests, one for males and one for females, were used to compare adult wing lengths between copepod absent and copepod present treatments. A Kruskal–Wallis test was also used to compare pupation day distributions between predator present and predator absent treatments. A nonparametric local regression method (“loess”, “ggplot2” package) was used to present the cumulative proportion of focal larvae pupated over time, by predator presence.

## 3. Results

Head capsule width measurements (mean ± SD = 0.59 ± 0.062 mm) of a subset of the focal mosquito larvae (n = 30) confirmed that they were third instars on the first day of exposure to copepod predation [55]. In total, 115 copepods (mean length ± SD = 1.75 ± 0.16 mm) were used throughout the experiment across the 70 predator treatment replicates.

Out of the 70 focal larvae that were not exposed to copepod predators, 97.1% emerged successfully as adults; one died in the larval stage, and one died in the pupal stage. Out of the 70 focal larvae that were exposed to copepod predators, 87.1% emerged successfully as adults; five died in the larval stage, and four died in the pupal stage. Results of a Pearson’s chi-squared test showed that the probability of successful adult emergence was higher in the absence of copepod predators (*p* = 0.0279). Three of the five focal larvae that died in the presence of a predator showed clear signs of melanisation in the abdominal region, most likely due to wounding from copepod attacks (Figure 2).

The percentage of early instar *Ae. albopictus* surviving each day was significantly lower in the presence of a copepod predator throughout the six days immediately following predator introduction (Table 1). No significant difference in early instar survival was observed on the last two days, when the number of remaining replicates was very low (Table 1).

*Ae. albopictus* adult wing length data were left-skewed among both males and females. Female wing lengths (median = 2.87 mm) were significantly larger than male wing lengths (median = 2.33 mm, Kruskal–Wallis rank sum test, *p* < 0.0001). However, there were no significant differences in male wing length (Kruskal–Wallis rank sum test, *p* = 0.6387) or in female wing length (Kruskal–Wallis rank sum test, *p* = 0.1769), due to copepod presence (Figure 3a).

Pupation day data were right-skewed among both males and females. Sex did not affect the day of pupation (Kruskal–Wallis rank sum test, geometric mean ± SD: males = 11.8 ± 1.1, females = 11.9 ± 1.1; *p* = 0.2607). In addition, neither male pupation day (geometric mean ± SD: copepod present = 11.8 ± 1.1, copepod absent = 11.7 ± 1.1; Kruskal–Wallis rank sum test, *p* = 0.7819), nor female pupation day (geometric mean ± SD: copepod present = 11.8 ± 1.1, copepod absent = 12.0 ± 1.1; Kruskal–Wallis rank sum test, *p* = 0.1580), differed with copepod presence (Figure 3b).

## 4. Discussion

Although third and fourth instar *Ae. albopictus* larvae are generally less vulnerable to copepod predators than first and second instars [19], some of the third instar, fourth instar, and pupal stage deaths observed in this study were likely due to *M. viridis* copepod attacks. Three focal individuals that died in the larval stage showed signs of melanisation (Figure 2), a response triggered locally by cuticular wounding that results in the accumulation of melanin, a brown-black pigment, at the wound site [58,59,60]. Melanisation is an energetically costly response that is likely to be influenced by nutritional status in mosquitoes [58,61,62]. Since the focal larvae in this study were provided with fish food ad libitum, it is unlikely that their immune responses were limited due to poor nutritional status. The location of the three melanisation sites in the abdominal region (Figure 2) is consistent with the findings of a previous study in which the larval abdomen was attacked more frequently by cyclopoid copepods than either the head/thorax region or the last body segment, containing the siphon [63].

While successfully emerging males were spread evenly between the predator and control treatments, fewer females emerged successfully from the predator treatment. These observations are consistent with those of a previous study showing that *Ae. albopictus* survivorship is skewed towards males in response to predation by *T. rutilus* [44]. A longer (five-week) semifield study found that the *Ae. albopictus* sex ratio was skewed towards females after extended exposure to cyclopoid copepod predation [45]. However, lower larval densities have been shown to produce lower proportions of males in *Ae. albopictus* rearing [64]. One possible explanation for this is that the increased nutrient availability for each larva at lower densities might better support the larger body size of females. Thus, lower larval density is likely to be the main cause of the female-dominated sex ratio that was previously observed after predation by copepods [45]. A small increase in mortality among late instar larvae due to predation, such as the 10% increase shown in this study, is unlikely to result in increased body size among surviving late instars, especially when nutrients are available at high levels, as is often the case in *Ae. albopictus* larval habitats observed in the field [65].

In order for cyclopoid copepods to be effective biocontrol agents, enough adult female copepods need to be applied in order to quickly eliminate first and second instar mosquito larvae, which are the most vulnerable stages to copepod predation [19]. In some cases of incomplete control, *Ae. albopictus* adults that developed in the presence of predators emerged larger than adults that developed in control conditions because the lower larval density produced by predators resulted in less intraspecific competition and greater per capita nutrition [44,45,46]. Early instar larvae may also benefit nutritionally from decomposing dead conspecifics [30]. First and second instar larvae are likely to benefit the most from increased nutrition because, on the basis of head capsule widths, most larval growth occurs between the first and third instar stages [55]. Under the scenario of incomplete biocontrol leading to the emergence of larger *Ae. albopictus* adults, it is important to consider that *Ae. albopictus* females do not avoid copepods during oviposition [19,45]. Therefore, the higher fecundity that is associated with larger female size [36,39] is likely to be strongly counteracted by copepod predation against newly hatched *Ae. albopictus* larvae of the next generation.

Among *Ae. albopictus* adults that emerged successfully in this experiment, there was a significant difference in wing length due to sex (Figure 3a), but there was not a significant difference in pupation day between sexes. *Ae. albopictus* males were previously observed to be 17–20% smaller than females [66]. Accordingly, the male median wing length in this study is 18.8% smaller than the female median wing length, and wing length is known to correlate positively with mass in *Ae. albopictus* [56]. Previous work showed that, while mass clearly differs due to sex, development time in *Ae. albopictus* is less sexually dimorphic [67]. There was no significant difference in wing length or pupation day due to *M. viridis* predation cues (Figure 3). Therefore, neither the greater reproductive success [36,37,38,39] and longer lifespans [40] observed among larger *Ae. albopictus*, nor the higher chance of dengue infection [42] and higher biting rates [43] observed among smaller female *Ae. albopictus* are likely to result directly from *M. viridis* predation cues. A similar lack of predator impact on mosquito size and development time was observed when *Toxorhynchites amboinensis* was tested against newly hatched *Ae. polynesiensis* in coconuts at three different larval densities [68]. In addition, a study spanning four generations of *Ae. albopictus* did not find any evidence of an evolutionary response to predator exposure in the larval stage [69].

## 5. Conclusions

We found evidence of lethal attacks on late instar larvae that resulted in a small, but significant, reduction in the probability of emergence. However, we did not find any evidence that experiencing *M. viridis* predation, both directly and on smaller conspecifics, affects size or development time of surviving *Ae. albopictus* adults at optimal nutritional availability. Previous work showed that *M. viridis* exhibits a type II functional response curve and a relatively high predation efficiency against *Ae. albopictus* prey at temperatures representative of UK larval habitats [50]. This study builds on recent work and further supports the use of *M. viridis* as a biocontrol agent for *Ae. albopictus* in South East England, at the northern edge of the vector’s expanding global range. Results indicate that late instar larvae exposed to this predator do not exhibit altered traits that could impact mosquito population growth or disease risk.

### Further Research

*Ae. albopictus* larvae are generally found in container habitats that have high levels of organic detritus [65]. In addition, ovipositing females prefer organic infusions to containers holding only water [70]; they also prefer high levels of detritus to low levels, and high-quality detritus to low quality [71]. Therefore, we found it most appropriate to test for sublethal effects under conditions of high nutrient availability. Further research is necessary to determine if the same results would be observed under low nutrient availability.

Although cyclopoid copepods have been shown to reduce mosquito populations in the field [12,13,14,15], it is likely that other components of “integrated vector control”, including chemical insecticides, will be employed concurrently with copepods against *Ae. albopictus* mosquitoes [72]. In addition, previous studies support the use of a bacterial biocontrol, *Bacillus thuringiensis* var. *israelensis* (*B.t.i.*), with cyclopoid copepods in standing water habitats [73,74]. *B.t.i.* is lethal to the third and fourth instar mosquito larvae that are less vulnerable to copepod predators [73], but it only provides short-term control, remaining effective against mosquitoes for three weeks or less [73,75,76]. Therefore, it is beneficial to have copepods introduced to the same habitats where *B.t.i.* is applied, so that the copepods can eliminate mosquito larvae that hatch after *B.t.i.* loses its toxicity [73]. In past experiments, *B.t.i.* did not have a toxic effect on *Macrocyclops albidus*, *Mesocyclops longisetus* or *Mesocyclops ruttneri* copepods [73,74], but further research is needed to evaluate the potential effects of *B.t.i.* on *M. viridis* copepods.

## Figures and Tables

**Figure 1 insects-13-00155-f001:**
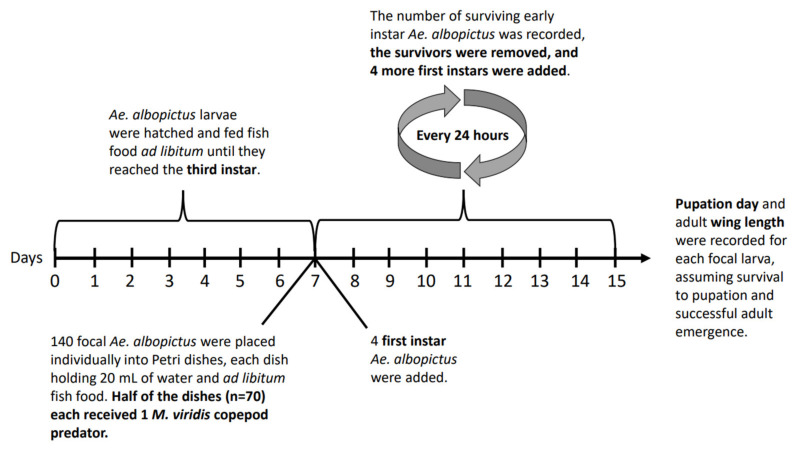
Experiment schedule.

**Figure 2 insects-13-00155-f002:**
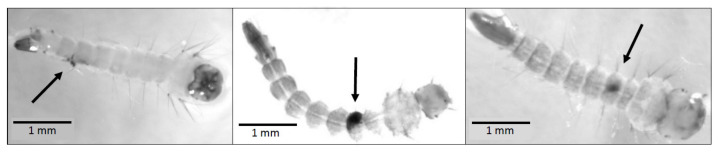
Evidence of melanisation in focal larvae that died in the presence of a copepod predator on the eighth, ninth, and tenth days of observation, respectively.

**Figure 3 insects-13-00155-f003:**
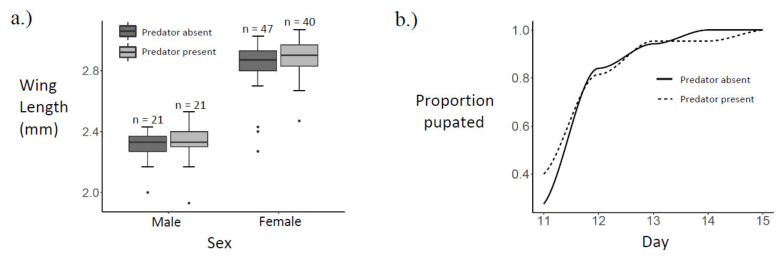
Size and development: (**a**) Boxplot of wing lengths by sex and predator presence; (**b**) cumulative proportion pupated by predator presence.

**Table 1 insects-13-00155-t001:** Early instar survival percentage by day and predator presence.

Day of Experiment	Predator Absent (Mean ± SE)	Number of Predator Absent Replicates	Predator Present (Mean ± SE)	Number of Predator Present Replicates	*p* Value ^1^
8	99.3 ± 0.5	70	6.8 ± 2.0	70	<0.0001
9	99.3 ± 0.5	70	30.1 ± 4.1	69	<0.0001
10	99.3 ± 0.5	70	47.8 ± 3.9	68	<0.0001
11	100 ± 0.0	70	42.9 ± 4.3	67	<0.0001
12	100 ± 0.0	42	55.9 ± 6.6	34	<0.0001
13	100 ± 0.0	12	58.3 ± 8.9	12	0.0007
14	100 ± 0.0	5	33.3 ± 33.3	3	0.1835
15	100	1	25.0 ± 14.4	3	NA

^1^*p* value corresponds to a Welch two-sample *t*-test for samples of unequal variance used to determine if there was a difference in early instar survival based on predator presence.

## Data Availability

All data that were collected during this study are publicly accessible from the Dryad Digital Repository: https://doi.org/10.5061/dryad.vmcvdncv7.

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
