# Peer review of "No Impact of Biocontrol Agent’s Predation Cues on Development Time or Size of Surviving Aedes albopictus under Optimal Nutritional Availability"

_insects, 2022, doi:10.3390/insects13020155_

Round 1
Reviewer 1 Report
In the manuscript "No impact of biocontrol agent’s predation cues on development time or size of surviving Aedes albopictus under optimal nutritional availability" authors showed, that the mosquitoes Ae. albopictus performance is not affected by the presence of predator.
The study is well designed, and the manuscript is well written. However, descriptions in several places seems to be too long. I suggest to shorten the Introduction section. Also, in some places, the language of the manuscript is complicated and there are long, subordinate complex sentences. Authors should review the manuscript and simplify the content when it is possible.
Also, authors should provides figures in better resolution.
In result section, line 211, authors should provide only one value – or raw data or percentage one.
In many place the description of the results is not necessarily long, eg. “mean = 0.59 mm, sd = 0.062 mm” can be expressed as “0.59±0.062 mm”. P-value can be replaced by just p.
I have no comments to the substantive side of the work.
Reviewer 2 Report
I have reviewed the manuscript titled “No impact of biocontrol agent’s predation cues on development time or size of surviving Aedes albopictus under optimal nutritional availability” by Russell and Cator. The manuscript presents the findings of an investigation to determine the potential role of a predator of immature mosquito stages to assist control of an exotic mosquito, Aedes albopictus. This was an interesting study investigating the role of alternative prey in mosquito biocontrol experiments. Studies investigating potential biological control agents of exotic mosquitoes is required, especially in regions of the world where invasive mosquitoes have the potential to change the local transmission risks of mosquito-borne pathogens of serious public health concern. Studies of this nature, whether demonstrating high effectiveness of biological agents in the control of mosquitoes, or ineffectiveness, should be noted to assist those organisations or institutions responsible for management of mosquitoes. As I understanding it, the authors have exposed immature stages of Aedes albopictus to the predatory copepod Megacyclops viridis. The hypothesis of the work is being that these predators may not cause the high rates of mortality in late stage immature mosquitoes stages compared to young stages. This work was undertaken by simultaneously exposing a single III stage larvae and II stage larvae to copepods under laboratory conditions in petri-dishes. Of those “focal larvae” reared with exposure to copepods but also with the presence of early stage larvae, approximately 87% survived to pupae and emerged with little evidence of sublethal effect based on adult wing length. During the larval rearing stages, daily survivorship varied from around 7% to 55%. As the authors did find some minor predation/attack of copepods on late stage larvae, it would be interesting to include a comment on the “real world” setting where there was not necessarily replacement of those younger larvae. When they were all gone, would the predation on older larvae have a far greater impact? For a mosquito that relies on fluctuation water levels for hatching of larvae, in many circumstances, there may be only a single cohort of larvae present in a water-filled container and not necessarily a mixed cohort. Does this mean that the results suggest copepods would have far greater impact on those larvae surviving to adulthood (i.e. without the alternative prey of young larvae)? I feel the most important issue to address in the manuscript is the conclusion based on these results. The authors state “we did not find any evidence that experiencing M. viridis predation, both directly and on smaller conspecifics, affects size or development time of surviving Ae. albopictus adults at optimal nutritional availability”. My interpretation of that is that these copepods are not necessarily an effective mosquito control agent, compared to larvicides or insect growth regulators, given the survivorship of later stage larvae. However, the authors also conclude that “…supports the use of M. viridis as a biocontrol agent for Ae. albopictus in the southeastern UK, at the northern edge of the vector’s expanding global range”. While the use of these copepods may have an impact of survivorship of early stage larvae, I’m not sure that this study supports their use for control (the manuscript's title suggests this). I think it is important that, if this is the recommendation of authors, there is some additional commentary in the discussion about the limitations of this approach where invasive mosquito control is of critical importance. I think it would be useful to include a brief comment on how the use of these copepods may compare to traditional larval mosquito control agents. This should include a comment on the operational challenges of rearing and distributing the copepods in an urban landscape, again in comparison to traditional most control agents. A minor comment. In a few places, the formatting could be revised to ensure captions aren’t listed on the last line of page, a different page to table (e.g. Line 231)Author Response
Please see the attachment.

Reviewer 3 Report
The authors report the influence of a local copepod species (M. viridis) on the survival, development time and adult body size of Ae. albopictus late instar larvae. They report that copepod attacks cause a negligible increase in late instar larval mortality, however copepod attacks/predator cues did not illicit significant differences in either development time or adult body size. The authors ultimately conclude that the use of M. viridis copepods is may be effective as biocontrol agents against Ae. albopictus in southeast England. The manuscript is well-written, and though local in focus, contributes to the knowledge base relative to biological control of invasive mosquito populations on a broader geographic scale.
Author Response
We thank Reviewer 3 for their positive feedback.